# Towards a Semantics-Based Recommendation System for Cultural Heritage Collections

Jiayu Li  and Antonis Bikakis *

Department of Information Studies, University College London, London WC1E 6BT, UK; jiayu.li.21@ucl.ac.uk
* Correspondence: a.bikakis@ucl.ac.uk

**Abstract:** While the use of semantic technologies is now commonplace in the cultural heritage sector and several semantically annotated cultural heritage datasets are publicly available, there are few examples of cultural portals that exploit these datasets and technologies to improve the experience of visitors to their online collections. Aiming to address this gap, this paper explores methods for semantics-based recommendations aimed at visitors to cultural portals who want to explore online collections. The proposed methods exploit the rich semantic metadata in a cultural heritage dataset and the capabilities of a graph database system to improve the accuracy of searches through the collection and the quality of the recommendations provided to the user. The methods were developed and tested with the Archive of the Art Textbooks of Elementary and Public Schools in the Japanese Colonial Period. However, they can easily be adapted to any cultural heritage collection dataset modelled in RDF.

**Keywords:** semantic technologies; recommendation systems; graph databases; digital cultural heritage

## 1. Introduction

Information management in the field of cultural heritage has been challenged by the diversity and heterogeneity in cultural data. Cultural materials are massive in quantity, diverse in topic, intricate in interconnection, and varying in form when stored digitally [1]. Semantic technologies, including the RDF data model and RDF-compatible ontology languages such as OWL, have been widely used to address these challenges, enabling the organisation and interoperability of distributed cultural heritage databases based on the meaning of data. Using machine-readable metadata, semantic technologies enable not only richer semantic descriptions of cultural heritage entities but also their connection with semantic relations, such as material type, geographical origin, the creator or current owner of an artefact, etc.

Several large cultural organisations have applied semantic technologies to their collections, including the British Museum, the Smithsonian American Art Museum, the Victoria and Albert Museum, Rijksmuseum, and many others, and several digital libraries that aggregate data from different collections have been developed using the same technologies, with Europeana [2] being the most notable example. In most of these cases, the focus has been on improving the organisation and management of cultural data, with cultural heritage professionals and researchers being the main beneficiaries, while little attention has been paid to services and applications for the end users. For example, visitors to most museum websites still have to use keyword-based search engines to explore and search through their collections, and they often have to go through the overwhelming search results and comprehend the culturally entangled information before finding the objects they are looking for. The existing semantically annotated cultural datasets have significant potential in supporting more sophisticated search engines and other user-end intelligent services, such as association discovery, personalisation, and recommendation [1]; however, such types of services can rarely be found in cultural portals.

Aiming to address this gap, this study develops recommendation methods making use of cultural heritage semantic datasets and the architecture of a recommender system for cultural heritage collections. The proposed methods aim to alleviate the users' burden when finding objects of interest in large collections by recommending objects based on their preferences and viewing behaviours. The methods were developed on top of a specific dataset, but they can easily be adapted to any semantic cultural dataset and reused by any cultural organisation that aims to improve the experience of visitors to their online collections. We implemented the proposed methods on top of the graph database platform Neo4j and its associated graph algorithms. By developing these recommendation methods and a recommender system architecture, we seek to achieve two main objectives:

- Demonstrate ways that graph database technology can leverage the semantically rich cultural heritage collection datasets that are currently available in several cultural heritage institutions and online digital libraries.
- Develop the foundation for a semantics-based recommendation system aimed at facilitating the exploration of cultural heritage collections in a personalised way, taking into account the user's individual preferences as explicitly stated by the users or derived from their viewing behaviours.

In the rest of this paper, we first present the background and related work of the study in Section 2. In Section 3, we present the recommendation methods and their implementation in a graph database; the dataset that we developed and used to test the methods; the architecture of the search and recommendation system for cultural heritage collections that uses the proposed methods; and their prospective evaluation. In the last section, we summarise and discuss the next steps of this work.

## 2. Background and Related Work

### 2.1. Semantic Technologies in Cultural Heritage

Cultural heritage refers to the legacy of physical artefacts and intangible attributes of a society inherited from the past, maintained in the present, and preserved for the benefit of the future [1,3]. Cultural heritage materials are mainly administrated by memory organisations, such as libraries, archives, and museums. Following the recommendations for Open Access to cultural heritage, also known as 'Open Galleries, Libraries, Archives, and Museums' (Open GLAM), many institutions worldwide have digitised their collections, organised the collection data into databases, and published them online. Cultural heritage data is massively heterogeneous in terms of topic, language, culture, data format, target audience, etc., while the items and collections can be "semantically extremely richly interlinked" [1,4]. Semantic technologies have become popular for facilitating data management in the cultural sector as they enhance the interoperability and reuse of cultural data and also enable the development of intelligent services on top of semantically annotated and interlinked data.

The Semantic Web has been discussed for over twenty years since Tim Berners-Lee et al. published the famous paper "The Semantic Web" in 2001 [5,6]. The paper describes an appealing future of the Web providing intelligent services based on machine-readable information. Compared to the current document-based Web, the Semantic Web adds a layer of machine-interpretable data, the metadata, which represents the meaning of resources and documents [1]. This layer of metadata is supported by the RDF (Resource Description Framework) data model and the OWL Web Ontology Language. Both rely on a graph-based data model that explicitly represents relationships between different web resources using triples of the form subject–predicate–object. Each web resource is assigned a unique URI (Universal Resource Identifier). Triples can be interlinked (creating a knowledge graph) to support wider data integration [7]. The aim of this model is to encode the semantic metadata of web resources and link web data to real-world entities in a manner that simulates human comprehension of a domain, facilitating communications between people and data management application systems [8]. An example of an RDF graph is depicted in Figure 1.

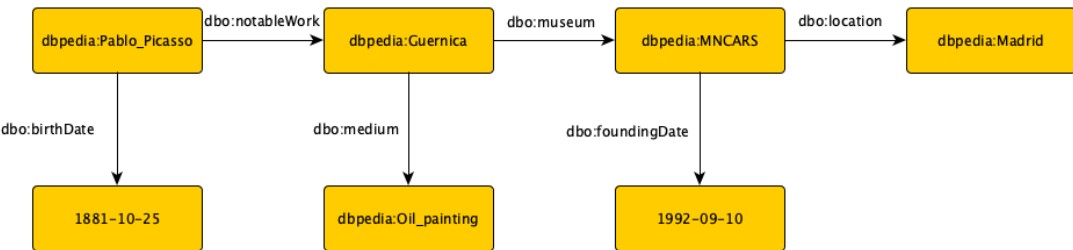

**Figure 1.** A set of RDF triples extracted from DBpedia [9].

Using the languages and technologies of the Semantic Web, semantic information portals support the management and provide access to semantically annotated data [10–12]. They facilitate the content publishers' work by collecting data from various web sources exploiting the homogeneous, ontology-based representations and interconnections of semantic metadata [13]. From the view of end-users, they provide intelligent services for accessing and visualising aggregated data, exploiting their underlying structure. During the last two decades, several semantic information portals have been established on top of cultural heritage collections, such as Europeana, the largest European digital library [14,15], and the Sampo portals focusing on various aspects of Finland's cultural heritage [16].

*2.2. Semantic Recommendation*

Semantic recommendation is one of the prevalent Semantic Web applications, although it has not been widely used in the cultural heritage domain. The origin of recommendation systems, also known as recommender systems, can be traced back to the early 1990s when one such system was developed to filter personal emails [17]. At present, personalised recommendation systems have become ubiquitous from online shopping to streaming services. The basic principle of recommendation is that there are significant dependencies between user- and item-centric activity. For example, a user who is interested in historical documentaries is likely to be attracted to educational programmes rather than action movies [18]. The dependencies can be learned in a data-driven manner, and the resulting model is used to make predictions for target users [18]. A widely acknowledged operational goal of recommendation systems is to recommend items that are relevant, while the novelty, serendipity, and diversity of the recommended items are also considered as potentially desired properties [18].

Recommendation systems are classified into three main types based on the recommendation method they use: collaborative, content-based, and knowledge-based. They work mainly with two kinds of data (1) the user–item interactions, such as ratings and browsing behaviours and (2) the profiles of users or items, which may include demographic information of users or search keywords [18,19]. Collaborative filtering models focus on the first type of data. They generate recommendations based on the observed user–item correlations. In a user-based collaborative system, a user is likely to be recommended an item that other users related to that user have expressed interest in. In item-based collaborative filtering methods, a user receives recommendations for the same type of items that the user has spent time on [18]. Content-based and knowledge-based recommendation systems similarly make use of the attributes of items. Content-based filtering analyses attributes of items that are related to users' recorded activities, such as rating and browsing, and provides recommendations based on the users' preferences. A knowledge-based system recommends items according to explicit specifications provided by the users beforehand, such as constraints on the item attributes or sample cases as targets or anchor points [18].

Semantically organised cultural heritage data suit recommendation systems, especially with content-based and knowledge-based methods [1,19]. There have been some attempts at semantic recommendation on cultural heritage portals. Ruotsalo and Hyvö-nen [20] proposed a method for determining the semantic relevance of annotations and used it to implement a knowledge-based recommendation system on the CultureSampo portal. The method calculates the ontological relevance of RDF resources based on the

number and type of data relationships encoded in an RDFS ontology, including class hierarchies and class–instance relations. Another example is the Art Recommender of the CHIP (Cultural Heritage Information Personalization) project on Rijksmuseum's InterAcrief database [21,22]. The system recommends art-related concepts based on user ratings of artwork and concepts from the underlying ontology used to model the museum's collections. Recommender systems empowered using semantic technologies have also been developed in other domains such as e-commerce [23], e-learning [24], and job recruitment [25].

### 2.3. Graph Databases for Semantic Recommendation

Graph databases have recently become an alternative and more efficient (compared to relational databases) way to store and manage highly relational data. They rely on a graph-based data model, called a property graph, with two main types of elements: vertices, which represent entities, connected by edges representing their relationships. Entities and relationships may also have attributes, which are used to further describe them. An example of one such graph, which represents the same information as the RDF graph shown in Figure 1, is depicted in Figure 2. Relationships are treated equally important as entities, which makes graph databases suitable for queries concerning paths and relationships, the centrality of nodes, and pattern-matching problems [26]. Such types of queries are often used by recommendation systems, which are based on representing and identifying associations among users and items [27]. Answering queries in a graph database usually involves accessing only a portion of the graph; this ensures a constant performance, which is not affected by the size of the database [27].

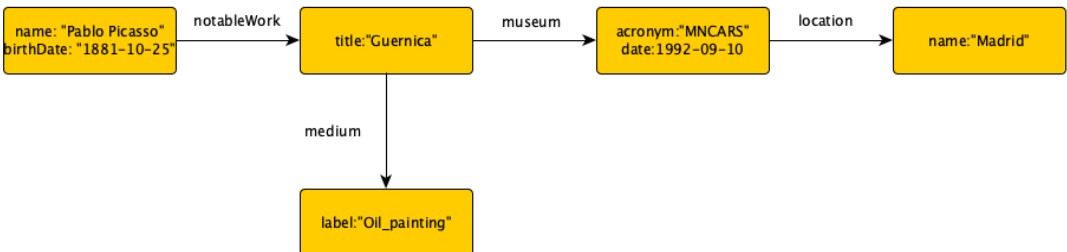

**Figure 2.** A property graph that is semantically equivalent to the RDF graph in Figure 1.

The similar graph-based structures of RDF datasets and graph databases enable their interoperability, making the use of graph databases to store, manage, and analyse RDF data a possible and very efficient solution. For example, it is possible to develop a semantic recommendation system that uses RDF to model and interchange data over the Web and graph databases to implement the recommendation algorithms, taking advantage of their advanced features with respect to access speed, storage efficiency, and application-specific algorithms and tools [28,29].

Recommendation systems powered using graph databases have been developed for various domains, such as for recommending movies [30], books [31], jobs [32], etc., while graphs have also been used as the underlying data structure for deep learning-based recommendation systems (see [33] for a relevant tutorial and related references). In the domain of cultural heritage, Sansonetti et al. [34] developed a hybrid recommendation system, the Cicero recommender, for suggesting cultural itineraries to users. The system combines two recommendation sub-systems, one based on a graph database storing information about the users' social networks obtained from Facebook, and the other based on an RDF database with a SPARQL endpoint used to retrieve related cultural heritage data from DBpedia and Europeana [34]. Another example is the system developed by [29], which integrates data from Wikidata, textual information from Wikipedia, and image-based information from Flickr and uses a graph database implemented in Neo4j to store and manage information about the cultural resources of Italy. Algorithms supported by Neo4j such as Page Rank are used to measure the popularity of Italian cities as cultural sightseeing

spots taking account of the connections between cities based on pictures taken by Flickr users [29].

## 3. Developing a Recommendation System for Cultural Heritage Collections

This section presents the design and implementation of recommendation methods for digital cultural heritage collections. It also presents the architecture of a hybrid recommendation system aimed at the users of cultural semantic information portals. The system combines strategies of content-based and knowledge filtering to recommend objects from a collection based on the semantic metadata of cultural objects and the users' interests, which are inferred from their search input and browsing behaviours. It generates a profile for each search, which is updated whilst the user interacts with the dataset by adding weight to the subjects or keywords the user is more interested in.

### 3.1. Description and Pre-Processing of a Collection Dataset

One of the aims of this study was to investigate why and how the semantics-based representation of cultural heritage resources enables more effective recommendation methods. It was crucial, therefore, to find and experiment with a well-formatted semantic dataset. While many institutions have developed semantic knowledge bases for their cultural heritage collections, most of them do not contain information about the features of the cultural objects. Some aggregated datasets that contain descriptions of multiple collections are even not homogeneous with respect to the metadata models or the languages they use. The Archive of the Art Textbooks of Elementary and Public Schools in the Japanese Colonial Period ("日帝殖民下台灣小公學校美術教科書暨影像數位典藏", AS-NTUE-School-Art-Textbooks) is a well-formatted dataset representing a single collection that meets our requirements. It contains semantic descriptions of 1697 pictures from the art textbooks used in Taiwan in the early 20th century, including subjects and keywords of the content of the pictures and other types of metadata. The dataset was originally established by the National Taipei University of Education Department of Digital Technology Design. It is now shared as part of the Linked Open Data Cloud provided by Academia Sinica Center for Digital Cultures under the Creative Commons Attribution CC 0 [35].

The dataset contains 59,224 RDF triples and was downloaded from Linked Open Data Cloud [35] in Turtle format. Each cultural heritage object has up to 21 fields of metadata in the form of either URI resource or text, mainly in Traditional Chinese. An example is presented in Table 1. The dataset contains 30 subjects that describe the content or genre of the pictures and 1386 keywords of diverse topics, either from DBpedia or in the form of text. Most objects are associated with one or more subjects and multiple keywords.

**Table 1.** Metadata of an object in AS-NTUE-School-Art-Textbooks. Chinese content was translated by the authors.

| Metadata Field (Explanation) | Metadata (Translation or URI information) |
| --- | --- |
| title | 汽車 (Cars); 自動車 (Automobiles) |
| label | 汽車 (Cars); 自動車 (Automobiles) |
| identifier | A01-00-008 |
| aat2427_produced_by (artistic media or techniques used to produce the picture) | http://vocab.getty.edu/aat/300011728 (fabricated chalk) |
| type (type of the object) | http://purl.org/dc/dcmitype/StillImage (still image); http://data.ascdc.tw/terms/Images (image); http://vocab.getty.edu/aat/300264387 (image) |

**Table 1.** *Cont.*

| Metadata Field (Explanation) | Metadata (Translation or URI information) |
|---|---|
| description | 畫面中為一藍色汽車，以鮮豔的淺藍色做為車身，再以黑線條描繪出形狀。日據時代時，日本人將西洋汽車引進台灣，進而成為貴族的代步工具，過去稱之為黑頭車。 (The picture shows a blue car with a bright light blue body and black lines to depict its shape. The Japanese introduced thewestern-style car to Taiwan in the Japanese Colonial Period, and it became the means of transport for the aristocracy and used to be known as the black-headed car.) |
| editor | / |
| author | / |
| printer | 中井利正（精版印刷株式會社代表者）(Toshimasa Nakai (Representative of Fine Printing Co., Ltd.)); 精版印刷株式會社 (Fine Printing Co.) |
| distributor | 台灣總督府 (Government-General of Taiwan) |
| dateCreated | 1935-02-28 |
| datePubllished | 1936-02-10 第三版發行 (Third edition issued on 1936-02-10); 昭和10年3月5日第一版發行 (First edition issued on March 5, Showa 10); 1935-03-05 第一版發行 (First edition issued on 1935-03-05); 昭和11年2月10日第三版發行 (Issued on the 10th of February, Showa 11, 3rd edition) |
| keywords | 汽車 (Cars); http://dbpedia.org/resource/Traffic (Traffic) |
| subject | 交通工具 (Transportation) |
| historyNote | / |
| collectedBy (individual or organisation who collected the object) | http://viaf.org/viaf/173080274 (National Central Library, Taiwan); http://viaf.org/viaf/66371957 (Yang, Mengzhe) |
| aggregatedCHO (link leading to the digital image of the object) | http://data.ascdc.tw/aggregation/uc/004577f6 |
| dataset (collection the object belongs to) | 聯合目錄 (Union Catalog); 日帝殖民下臺灣小公學校美術教科書暨影像數位典藏 (Archive of the Art Textbooks of Elementary and Public Schools in the Japanese Colonial Period) |
| relation (related resource) | 日帝殖民下台灣小公學校美術教科書暨影像數位典藏 (Archive of the Art Textbooks of Elementary and Public Schools in the Japanese Colonial Period) |
| rights | 保護期間已屆滿，為不受任何權利保護之公共財。(The period of protection has expired and the public property is not protected by any rights.) |
| uri | http://data.ascdc.tw/uc/art/ntue/004577f6 |

We used Neo4j (Neo4j Desktop 1.4.15, Database 4.4.8) to store and manage the collection metadata and create the underlying database for the recommendation system. Neo4j is a popular graph database management system implemented in Java. It does not only model data using graphs but also uses an underlying storage designed specifically for the storage and management of graphs. This makes it very efficient and highly scalable with respect to the retrieval, management, and analysis of connected data [36]. Neo4j uses property graphs to model data. Nodes in such a graph represent entities, and each edge represents a one-to-one relationship between the two nodes it connects. Labels on nodes represent the types or roles of the entities. Node/edge properties (in the form of key–value pairs) represent the attributes or additional metadata of the corresponding entity or relationship. Figure 3 shows an example schema of a graph database, where nodes are grouped by their labels. Neo4j allows users to import RDF data into a graph database and provides several graph algorithms for analysing the properties of property graphs. These two features of Neo4j were the main reasons for choosing it as the underlying database system for our recommendation system.

Before importing the dataset into Neo4j, we installed the Neosemantics plugin (4.4.0.1), which enables the manipulation of RDF datasets. We used its default configuration except for two settings. To produce a neat graph, prefixes of namespaces were set as "ignored" in the graph presentation. Furthermore, since some metadata fields have several values, while Neo4j allows only one value for each node property, the value format of node properties was set to "arrays" to retain all values.

The AS-NTUE-School-Art-Textbooks dataset was then imported into Neo4j. Although our methods only use part of the available data, we decided to import the entire dataset to deal with potential issues caused by interlinks of metadata from different fields and also to make it possible to support further recommendation methods in the future. As set by default, subjects of triples are mapped to nodes, while predicates of triples are mapped to node attributes if the objects of the triples are data values, or to relationships if the objects are resources [37]. Each node is assigned one or more labels, which correspond to the classes that the entity represented by the node belongs to. Two classes that are common in the AS-NTUE-School-Art-Textbooks dataset are AscdcProvidedCHO and AscdcAggregation. The former represents cultural heritage objects, while the latter contains aggregator links that connect the objects with the corresponding digital images. Two types of image resources are connected to each object: the image file, which is represented by an AscdcWebResource node connected to the node representing the object with an isShownBy relationship, and the webpage where the image is located, which is connected to the same node with an isShownAt relationship. The following Turtle code is a set of triples from the original RDF dataset, which demonstrates the use of the entities and relationships discussed above, while Figure 3 depicts the schema of the graph database we created.

<http://data.ascdc.tw/uc/art/ntue/004577f2> rdf:type  ns1:AscdcProvidedCHO.
<http://data.ascdc.tw/aggregation/uc/004577f2> rdf:type ns1:AscdcAggregation.
<http://data.ascdc.tw/aggregation/uc/004577f2> ns3:aggregatedCHO
<http://data.ascdc.tw/uc/art/ntue/004577f2>.
<http://data.ascdc.tw/aggregation/uc/004577f2> ns3:isShownAt
<http://catalog.digitalarchives.tw/item/00/45/77/f2.html>.
<http://data.ascdc.tw/aggregation/uc/004577f2> ns3:isShownBy
<http://image.digitalarchives.tw/ImageCache/00/50/fc/c9.jpg>.

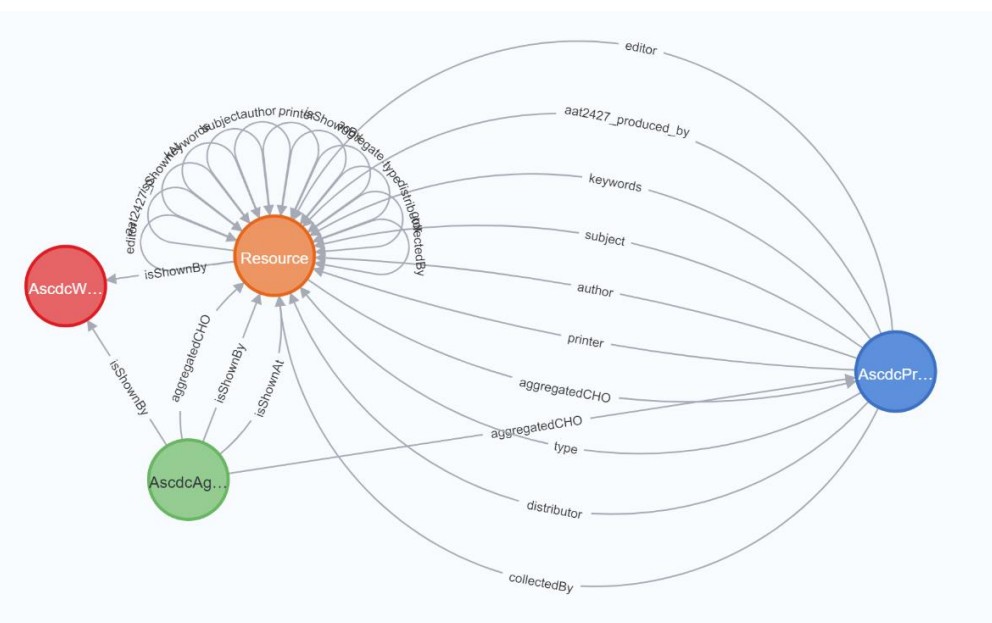

**Figure 3.** Schema showing the graph database we created from the AS-NTUE-School-Art-Textbooks dataset. Screenshot from Neo4j. (Node type: red: AscdcWebResource; blue: AscdcProvidedCHO; green: A3cdcWebResource; orange: Resource).

Moreover, each node that corresponds to a resource (entity with a URI) in the RDF dataset (including the entities discussed above) was also assigned the Resource label. Nodes with only the Resource label contain significant amounts of metadata, the type or role of which is represented by the different types of relationships directed to Resources nodes, such as distributor, printer, etc., as shown in Figure 3 and Table 2. Since Resource is a label that is assigned to all nodes, any relationship connecting nodes of any type appears in the schema of the graph database and also as a relationship starting from and ending to a Resource node. Figure 3 presents this group of relationships, whose type (indicated by the overlapping labels) includes all the relationship types as listed in Table 2.

**Table 2.** Relationship types and numbers in the graph database.

| Relationship Type | Amount |
|---|---|
| type | 5091 |
| author | 423 |
| editor | 21 |
| printer | 395 |
| subject | 1167 |
| keywords | 1434 |
| isShownAt | 1697 |
| isShownBy | 1697 |
| CollectedBy | 1721 |
| Distributor | 306 |
| aggregatedCHO | 1697 |
| aat2427_produced_by | 1326 |

Any metadata that has the form of a data value is assigned to an attribute of the corresponding node. Table 3 lists the attributes that appear in each type of node. The occurrence of some metadata fields as both node attributes and relationship types (e.g., editor and distributor) indicates that those fields contain both data values and resource URIs.

**Table 3.** Node labels and attributes in the graph database.

| Node Type (Label) | Amount | Associated Node Attribute |
|---|---|---|
| AscdcAggregation | 1697 | uri |
| AscdcProvidedCHO | 1697 | title, uri, subject, rights, relation, printer, label, keywords, identifier, historyNote, editor, distributor, description, datePublished, dateCreated, dataset, collectedBy, author |
| AscdcWebResource | 1697 | uri, license |
| Resource | 7340 | uri |

### 3.2. Identifying the Users' Interests

The system we developed targets users who have specific search goals but cannot pinpoint certain objects, which is common among cultural heritage experts [38] and could also happen to common users. For example, a curator might seek objects of specific themes in large collections they are not familiar with or a researcher might look for objects with features similar to those they already found in a collection [38]. While some cultural portals provide keyword searching and feature filters, these functions cannot meet the needs of every user, especially when they search for objects related to very specific topics that are not covered by the filters or when users have little knowledge of the terms used to describe the cultural heritage objects in the metadata.

The system identifies users' interests in two ways. At the beginning of a search, we assume that users will input information they think is relevant to their search goal, which

serves as the initial profile of the search. We assume that the searching interface provides subject and keyword lists and that the user can select and/or exclude items from the lists. When the system receives input information, it adds weight to the corresponding metadata so that the objects that match or are close to users' interests are prioritised over other objects.

Users' interests are also inferred from the objects they have viewed. Users usually browse several objects before they find the ones they are most interested in. The viewed object, if not the perfect match, is probably similar to what the users are looking for in some ways. Every time the user views an object, the system considers the object as part of the user's interests and adds weight to the subjects and keywords it relates to, unless the user marks the object as irrelevant, which we assume the viewing interface allows users to do. The change in weights will affect the recommendations the system provides for users at later stages. This approach aims to identify object features that users may be interested in but are not able to indicate using the subject and keyword lists provided by the system.

In the AS-NTUE-School-Art-Textbooks dataset, the system produces recommendations based on two metadata types: subject and keywords. The two metadata types are utilised mainly because they manifest the advantage of semantic data retrieval over text-based approaches. If a user looks for artefacts created by a certain artist, searching for the name of the artist with text could be as effective as searching for the semantic metadata. But a text-based programme can hardly process non-textual information, such as contents of images and 3D models, which requires additional annotations, such as subject and keyword metadata. Furthermore, as different terms (synonyms) may be used to describe the same feature of cultural objects, users may fail in finding the object they want only because they do not search for the specific descriptive term used in the dataset. A recommendation system can help users take advantage of the semantic annotations of objects effectively without having to learn the specific terms that are used in the dataset to describe the collection. Furthermore, the URIs of the terms that are used to describe subjects or keywords in the dataset accurately indicate their meaning (for example, "http://dbpedia.org/resource/Japanese_painting" obviously refers to Japanese paintings) and, more importantly, point to web pages that describe their meaning in detail.

As discussed in the previous section, the subject and keywords metadata fields include two types of data: web resources identified by a URI, which are mainly English entries from DBpedia, and literal data, which are traditional Chinese texts. The amount of metadata for each type and language is listed in Table 4. We should note that, despite their large amount, most Chinese keywords are used to describe only a small number of objects. We decided to ignore such terms in the recommendation algorithms we implemented because they have a significant impact on the complexity of the algorithms without contributing much to their performance. In the searching and viewing process, we assume that the cultural portal interface copes with the language issue and that users can use information in both languages freely.

**Table 4.** Number of subjects and keywords.

| Node | Amount | Node Property | Amount |
|---|---|---|---|
| English subject | 9 | Chinese subject | 21 |
| English keyword | 357 | Chinese keyword (ignored) | 1029 |

### 3.3. System Architecture and Recommendation Algorithms

The recommendation system consists of two sub-systems: the Search Module, which implements the search process, and the Recommendation Module, which implements the viewing process and creates recommendations. Our main focus is the Recommendation Module. While considering that the recommendations rely on both the users' interaction with the search system (by selecting topics of interest before the search starts) and the objects that the users chose to view from the search results, we also developed the Search

Module to receive user's input information and to generate suitable "hook objects" in the search results.

The recommendation system uses a collaborative approach, combining content-based and knowledge-based filtering methods. It produces recommendations according to users' selection or exclusion of keywords/subjects and also takes account of users' potential interests inferred from the objects they view. Working on top of a graph database with nodes representing collection items or keywords/subjects, the system uses the Node Similarity algorithm to compute the similarity among items based on their connections with the keyword/subject nodes and recommends items that are most similar to those that the user has already shown an interest in.

Users interact with the recommendation system as shown in Figure 4. They select and/or exclude subjects/keywords to start a search, and the Search Module generates a list of cultural objects that match their interests. The Search Module aims to provide objects which, in addition to meeting the users' requirements, connect to as many subjects and keywords as possible. When users view these objects, a wide variety of topics will be marked as users' interests, so that the recommendations will have the widest possible coverage, which increases the probability the system meets the users' needs.

If the user stops looking for items after viewing the objects returned in the search results, the process will end. In most cases, the user will continue viewing more objects, disclosing more information about their preferences. The Recommendation Module recommends objects which are similar to the last object the user viewed, as it could be the closest to the user's target at the current stage. The procedures related to the Recommendation Module can be repeated several times until the user stops viewing. The more times it is repeated, the more accurately the recommended objects are supposed to match the user's interests.

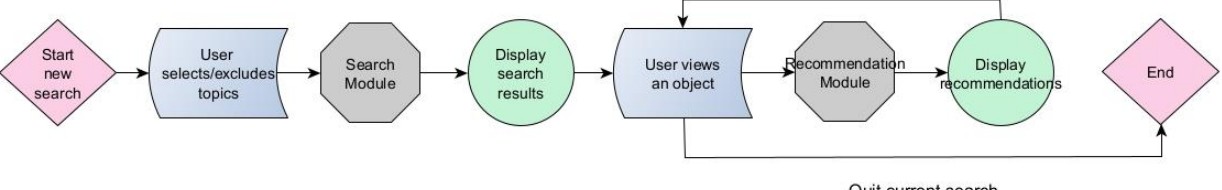

**Figure 4.** Process of user interaction with the recommendation system.

As Figure 5 shows, the recommendation system consists of four parts: Preparation, Search Module, Recommendation Module, and End of Search. The four parts cooperate to generate search results and recommendations and ensure that search activities are not affected by profiles that they are not associated with. The recommendation methods used by the system were implemented in Neo4j using the graph algorithms from the Graph Data Science Library (2.1.6) plugin of Neo4j Desktop.

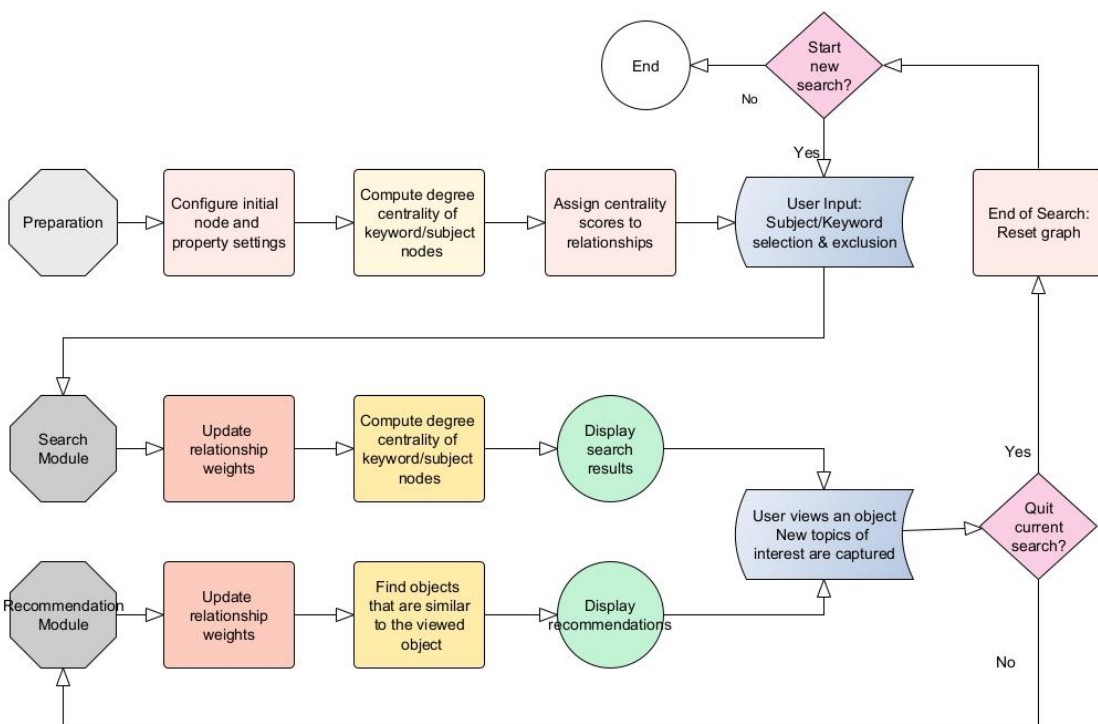

**Figure 5.** Recommendation system workflow.

(1)    Preparation

Before interacting with users, the Preparation Module sets a series of nodes, properties, and values to support the whole system. To support the Search and Recommendation Modules, which are based on identifying sets of objects with common subjects or keywords, the Preparation Module analyses the graph to assign weights to the relationships between objects and their associated subjects or keywords. Steps 1.1 and 1.2 (described in detail below) reformat the graph and create a graph projection that the Neo4j algorithms can be applied to, step 1.3 measures the centrality of subjects and keywords nodes, and then step 1.4 produces weights for subject/keyword relationships of object nodes based on the results of step 1.3. These settings are configured only once and are applied to each search activity.

(1.1)  Node and property settings

When importing data from the RDF dataset into Neo4j, the subjects associated with each object take the form of a value array, with each element in the array corresponding to a different subject. In order to be able to use Neo4j's algorithms (which exploit nodes or single-value node attributes rather than value-array attributes), we used the following Cypher statement to extract each subject from the value arrays, convert it into a separate node, and connect it with a subject relationship to the object it is associated with. We also assigned a uri property to each such node, which takes as a value either the URI of the subject or its textual description. In this way, we achieved a uniform representation of all subjects so that they are all treated in the same way by the Neo4j algorithms.

```
MATCH (n:AscdcProvidedCHO)
UNWIND n.subject AS w
MERGE (a:Resource {uri:w})
MERGE (n)-[:subject]->(a)
```

With the following Cypher statement, we added the "Topic" label to all nodes representing a subject or a keyword. We did this to differentiate from all other nodes that have a generic "Resource" label; this was essential for the recommendation algorithms.

```
MATCH (:AscdcProvidedCHO)-[:keywords|subject]-(a:Resource)
SET a:Topic
```

We also added a node property called "viewed" to every Topic node with the default value of 0. This is used to count the times a user views an object related to a specific subject or keyword and is also used as a weight by the recommendation algorithms.

```
MATCH (a: Topic)
SET a.viewed = 0
```

(1.2) Graph projection for subject/keyword centrality measurement

Graph projection is a materialised view over the stored graph, which is stored in the computer memory, allowing one to run algorithms over the graph (or a portion of it) quickly and efficiently [39]. We created graph projections containing only nodes with the label Topic (representing subjects or keywords) or AscdcProvidedCHO (representing objects of the collection), using the following Cypher statement.

```
CALL gds.graph.project.cypher(
    'projection',
    'MATCH (n) WHERE n:AscdcProvidedCHO OR n:Topic
        RETURN id(n) AS id',
    'MATCH (n:AscdcProvidedCHO)-[r:subject|keywords]->(m:Topic)
        RETURN id(n) AS source, id(m) AS target',
    {validateRelationships: false})
```

(1.3) Centrality of subject/keyword: Degree Centrality

The centrality of all Topic nodes is measured before the Search Module starts. Centrality algorithms are used to determine the importance of nodes in a network according to the number, weight, and other attributes of their relationships with other nodes [40]. Here, the centrality of a Topic node is computed based on the number of objects the Topic node is connected to, and it is used to rank objects in the search results of the Search Module. At the beginning of a search, users are likely to view objects from the top of the list of search results. For this reason, we configured the search module to prioritise objects that are associated with many subjects or keywords. In this way, we broadened the range of recommendations given to the user in the Recommendation Module and enhanced their diversity.

Among the centrality algorithms supported by Neo4j, four algorithms satisfy the needs of this procedure: Page Rank, Article Rank, Eigenvector Centrality, and Degree Centrality. We decided to use Degree Centrality because it is the only algorithm that also meets the requirements of centrality computation in the Search Module, so as to keep the algorithms consistent within the entire system. Degree Centrality computes the sum of incoming or outgoing (or both) relationships of a node. The following Cypher statement computes the number of relationships coming out of Topic nodes in the graph projection created in step 1.2. The algorithm is implemented in "write" mode; this results in the creation of a "centrality" property in every Topic node, with the centrality score assigned as its value.

```
CALL gds.degree.write(
    'projection',
{writeProperty: 'centrality', orientation:"REVERSE"})
```

(1.4) Subject/keyword weight preparation

To promote objects associated with high-centrality subjects/keywords, the centrality score yielded from the previous step is set as the initial relationship weight. The centrality

scores for this dataset vary from 0 to 357. To make the system easily adaptable to other datasets or to changes in the same dataset, we decided to normalise these scores before assigning them as weights. To do this, we used the Rescaling (min–max normalisation) method [41] to rescale the centrality score range in [0, 1] using the formula below (where *x* represents the original centrality score of a Topic node, *min*(*x*) and *max*(*x*) represent the minimum and maximum, respectively, centrality scores among all Topic nodes, and *x*′ represents the normalised score).

$$x = \frac{x - min(x)}{max(x) - min(x)} \tag{1}$$

We used the following Cypher statement to execute Degree Centrality in "stats" mode in order to retrieve the maximum and minimum centrality scores and then normalise the centrality scores as described above.

```
CALL gds.degree.stats(
    'projection',
    {orientation:"REVERSE"})
YIELD centralityDistribution
WITH centralityDistribution.min AS min, centralityDistribution.max AS max
MATCH (a:Topic)
WHERE exists (a.centrality)
SET a.centrality = (a.centrality − min)/(max − min)
```

After the normalisation, we assigned the centrality value of each Topic node as the weight of every relationship connecting the Topic node with an object. The reason for doing so is that the algorithms we used in the Recommendation Module do not take into account the property values of nodes. In order to save computation memory and avoid confusion with projections in the following steps, we deleted the projection. Figure 6 presents the topics with the top centrality values and their centrality scores, the topic "攝影寫真 [Photographic portrait]" being the one associated with the most items.

```
MATCH (a:Topic)
WHERE exists (a.centrality)
WITH a
MATCH (a)-[r:subject|keywords]-(:AscdcProvidedCHO)
SET r.weight = a.centrality

CALL gds.graph.drop ('projection')
```

| a.centrality | a.uri |
|---|---|
| 0.9999945504621753 | "攝影寫真" |
| 0.8907514483108452 | "人物群像" |
| 0.8655415016605383 | "靜物" |
| 0.812320503176557 | "自然風景" |
| 0.7759061357927802 | "http://dbpedia.org/resource/Japanese_people" |
| 0.7226851373087989 | "http://dbpedia.org/resource/Household_goods" |
| 0.6386519818077758 | "建築設施" |
| 0.6246464558909386 | "http://dbpedia.org/resource/Japanese_painting" |

**Figure 6.** Topics with top centrality values and their centrality scores. Screenshot from Neo4j. The topics listed in the figure are: 1. Photographic portrait, 2. Group portrait, 3. Still life, 4. Natural scenery, 5. Japanese people, 6. Household goods, 7. Buildings and installations, 8. Japanese painting.

(2)   Search Module

After the system is configured as described above, users are invited to interact with it. As an example, we assume that the user selects subjects/keywords T1 and T2 and excludes T3. In the Search Module, step 2.1 searches for subjects/keywords T1 and T2 and adds weight to their relationships to object nodes in order to promote these objects in the list of results the user will receive, while step 2.2 excludes objects with T3 and prepares graph projection for the following action. Step 2.3 measures the centrality of object nodes based on the relationship weights assigned in steps 1.4 and 2.1 and then returns the list of search results to the user.

(2.1) Weight added to the selected subjects/keywords

When the user selects subjects/keywords, the system adds additional weight to the relationships connecting objects to the corresponding Topic nodes. The value of the additional weight is set as the maximum number of topics an object can be associated with, which is 10 in the example dataset. The reason for using this number is described below.

The Search Module generates search results based on the weighted Degree Centrality of objects (described in step 2.3), which is affected by three factors: the centrality score of subjects/keywords (see step 1.4); the additional weight assigned to relationships after the user selects subjects/keywords; and the centrality score of objects. We want the system to prioritise the user's selection of subjects or keywords over the centrality scores of objects. In this way, an object that is associated with all the subjects/keywords the user selects will appear higher than any object that may be connected to many more subjects/keywords in total but to only some of the ones that the user selected. Assigning the maximum number of subjects/keywords that an object can be associated with as an additional weight in the relationships connecting objects to the selected subjects/keywords achieves our purpose.

The following Cypher statement retrieves the maximum number of subjects/keywords an object can be associated with, assigns it to the node viewed property of the Topic nodes that represent the subjects/keywords selected by the user, and updates (as described above) the relationship weight. The reason for assigning this value to the viewed property is that we want to use it as a base score for the selected Topic nodes. The value of the viewed property of each Topic node will then be incremented every time the user views an object associated with this node.

```
MATCH (a)-[:subject|keywords]-(n:AscdcProvidedCHO)
WITH n AS object, count(a) AS w
ORDER BY w DESC LIMIT 1
MATCH (a)-[r:subject|keywords]-(:AscdcProvidedCHO)
WHERE a.uri = "T1" OR a.uri = "T2"
SET a.viewed = w
WITH a, r
SET r.weight = a.viewed + a.centrality
```

(2.2) Graph projection for object centrality computation

In order to use the updated relationship weights to compute the centrality of object nodes, we created a new graph projection. The new projection is similar to the previous one but excludes the T3 nodes as the user requested, so any object related to T3 will not be included in the search results.

```
CALL gds.graph.project.cypher(
    'projection',
    'MATCH (n) WHERE n:AscdcProvidedCHO OR n:Topic
        AND NOT n.uri = "T3"
        RETURN id(n) AS id',
    'MATCH (n:AscdcProvidedCHO)-[r:subject|keywords]->(m:Topic)
    RETURN id(n) AS source, id(m) AS target, r.weight AS weight',
{validateRelationships: false})
```

(2.3) Centrality of object: weighted Degree Centrality

The Search Module presents the search results ranked by the weighted Degree Centrality of the objects, which depends on the weights of the relationships of each object and the centrality of the object nodes.

Like step 1.3, the Page Rank, Article Rank, Eigenvector Centrality, and Degree Centrality algorithms were considered for computing centrality. We selected Degree Centrality as it is the only algorithm among the four that considers both incoming and outgoing relationships. As shown in steps 1.2 and 2.2, subject/keywords relationships point from AscdcProvidedCHO (object) to Topic nodes. The other three algorithms compute the centrality of nodes based only on the incoming relationships; therefore, they would not be able to compute the centrality of object nodes taking into account their subject/keywords relationships. Another reason for choosing Degree Centrality is that it is the most sensitive to the relationship weight among all these algorithms. Experiments conducted using all four algorithms show that the number of relationships contributes much more than the relationship weight to the centrality score of each object in the other three algorithms. In our system, this would mean that an object associated with a small number of subjects/keywords will never appear at the top of the list of results, even if these subjects/keywords are exactly the ones selected by the user.

The following Cypher statement computes the weighted Degree Centrality of object nodes. Executed in "stream" mode, the algorithm returns the result directly, ranking the results from the highest weighted centrality to the lowest one. The result provides the URI of each object, which corresponds to the URL of a web page that presents the object. On a cultural portal, users would be able to view the objects they are interested in by clicking this URI. We chose to display the results in chunks of 30 considering the limited number of results users normally get on a single page and also to save computation memory in Neo4j. The "SKIP 30" clause (which is commented out in the code) is used to present the second chunk of results; for the third chunk, we used "SKIP 60", and so on.

```
CALL gds.degree.stream(
    'projection',
    {relationshipWeightProperty: 'weight'}
)
YIELD nodeId, score
RETURN gds.util.asNode(nodeId).uri AS object, score
ORDER BY score DESC
//SKIP 30
LIMIT 30
```

Figure 7 shows an example list of the results of this step. In the example search process, we assumed the user is interested in topics "花卉 [Flora]" and "Japanese paintings" (T1 and T2, respectively), but is not interested in items about "Bird" (T3). The Search Module provides a list of 30 results ranked by their centrality scores.

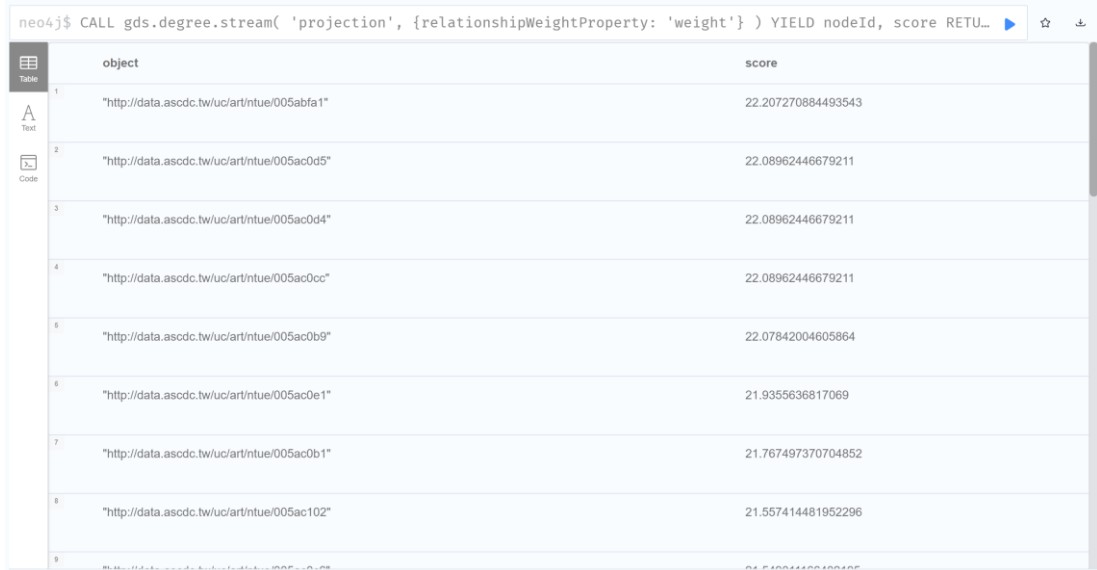

**Figure 7.** Part of the example result of step 2.3. Screenshot from Neo4j.

(3)　　Recommendation Module

While users view objects from the list of results, the system captures their potential interests. Every time a user views an object, the relationship weight of the subject/keywords relationships of the object will increase, unless the user marks the object as irrelevant, which we assume the interface of the cultural portal would allow the users to perform, and in which case, the related relationship weight does not change. As an example, we assume that the user views the object with uri = "OB1" and continues interacting with the system.

In the Recommendation Module, step 3.1 adds weight to subject/keyword relationships of OB1 to indicate the user's potential interest. After the preparatory work of graph projection in step 3.2, step 3.3 searches for objects similar to OB1 in terms of the related subject/keyword, while taking account of the additional weight accumulated in the process, and then returns the list of recommended objects to the user.

(3.1)　Weight added to subjects/keywords of the viewed object

When an object is viewed, the system will increase the value of the viewed property of the Topic nodes connected to the object and the weight of the corresponding relationships. We chose to increase such values only by 1, as viewing an object does not represent the explicit interest in the related subjects/keywords that the user's selection of subjects/keywords indicates at the beginning of their interaction with the system, and

should not, therefore, significantly affect the recommendations. We remind the reader that the initial values of viewed property of Topic nodes and the relationship weight of the subject/keywords relationships were set in step 2.1 based on the user's selection of subjects/keywords.

```
MATCH (a{uri:"OB1"})-[r:subject|keywords]-(b)
SET b.viewed = b.viewed + 1
WITH b, r
SET r.weight = b.centrality + b.viewed
```

(3.2) Graph projection for object similarity measurement

Similar to step 2.2, we deleted the old projection and created a new one to account for the updated relationship weights. T3 is excluded from the projected graph, as the user requested.

```
CALL gds.graph.drop ('projection')

CALL gds.graph.project.cypher(
    'projection',
    'MATCH (n) WHERE n:AscdcProvidedCHO OR n:Resource
        AND NOT n.uri = "T3"
        RETURN id(n) AS id',
    'MATCH (n:AscdcProvidedCHO)-[r:subject|keywords]->(m:Resource)
     RETURN id(n) AS source, id(m) AS target, r.weight AS weight',
{validateRelationships: false})
```

(3.3) Similarity of object: Filtered Node Similarity

The Recommendation Module aims to find and recommend objects that are similar to the last object the user viewed. For this purpose, we used the Node Similarity algorithm, which compares two nodes based on the number of nodes they are connected to, using the Jaccard metric. In the formula below, which is used to compute the Jaccard metric, $A$ and $B$ represent the set of nodes for each of the two nodes we compare.

$$J\left(A,B\right) = \frac{|A \cap B|}{|A \cup B|} = \frac{|A \cap B|}{|A| + |B| - |A \cap B|} \tag{2}$$

The filtered version of this algorithm (Filtered Node Similarity) allowed us to compute only the similarities between "OB1", which is the object that the user viewed (specified by the sourceNodeFilter parameter of the algorithm), and all other objects. The topK parameter indicates the number of results we want the algorithm to return, which is 10 by default. Given the size of the collection we experimented with, we set this parameter to 100. In stream mode, the algorithm returns results ranked by the similarity scores (from the highest to the lowest). With the two last clauses of the Cypher statement that implements the algorithm (shown below), we configured the algorithm to present the results in chunks of 10. We assumed that this is a parameter that the cultural portal may ask the user to provide as input.

```
MATCH (a{uri:"OB1"})
WITH id(a) AS viewedNode
CALL gds.alpha.nodeSimilarity.filtered.stream(
'projection',
{relationshipWeightProperty: 'weight',
sourceNodeFilter: viewedNode, topK:100})
YIELD node2, similarity
RETURN gds.util.asNode(node2).uri AS recommendations, similarity
ORDER BY similarity DESC
//SKIP 10
LIMIT 10
```

Figure 8 presents an example list of recommendations. Following the search results produced in step 2.3 (as shown in Figure 7), we assumed the user chooses to view the first item in the list (http://data.ascdc.tw/uc/art/ntue/005abfa1). We assumed that the user's interests (in "花卉 [Flora]" and "Japanese paintings" but not in "Bird") remain the same. The Recommendation Module provides a list of 10 results ranked by their similarity scores.

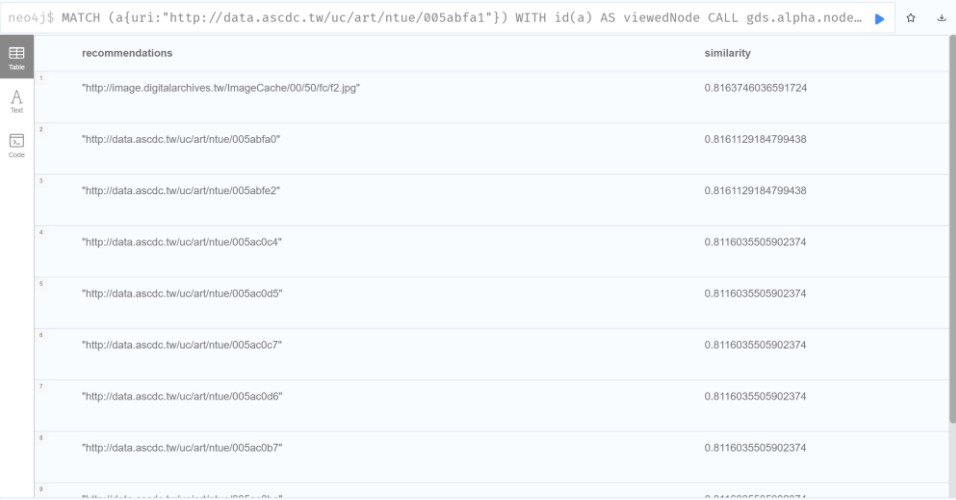

**Figure 8.** Part of the example result of step 3.3. Screenshot from Neo4j.

After receiving the recommendations, users can view one of them (by clicking their URI), which triggers more recommendations (unless they mark the object as irrelevant), until they stop the current search.

(4)　End of search

When the user stops searching and viewing, the last graph projection will be deleted. The value of the viewed property of AscdcProvidedCHO nodes (object) and the relationship weight of subject/keywords relationships will be cleared to accommodate a new search.

```
CALL gds.graph.drop ('projection')

MATCH (a:AscdcProvidedCHO)-[r:subject|keywords]-(b)
SET a.viewed = 0, r.weight = b.centrality
```

As Figure 5 demonstrates, when a new search starts, the preparation procedure will not be performed, and the system will start with the Search Module.

### 3.4. Evaluation

Evaluation plays a crucial role in the development of recommendation systems. It usually focuses on the accuracy of the recommendations, using metrics such as precision

and recall, which measure the quality and quantity of recommendations, respectively [19]. Depending on the purpose and context of the recommendations, evaluation goals can also include coverage, trust, novelty, serendipity, diversity, etc. [16]. Due to time restrictions and insufficient resources, we were not able to conduct a user-based evaluation of the proposed recommendation algorithms. In this section, we describe how the proposed algorithms can be evaluated in the future. For the Search Module, the evaluation concentrates on assessing the extent to which the search results match the user's interests (precision) and the variety of topics they are related to (diversity). For the Recommendation Module, in addition to precision, efficiency should also be considered, as the system is designed to capture the users' additional topics of interest, which are reflected by their viewing behaviours, and locate target objects in more than one round of interaction. The assessment of recall is also important for both modules, although it is difficult to achieve with the example dataset because the available semantic metadata does not describe all the visual features of some objects.

The following sections introduce the evaluation process, including the aspects of evaluation for each module and the related measures. Most procedures concern analysis of results produced using the proposed system; a few procedures require additional actions implemented with Cypher statements, which are presented in the Appendix A.

(1)   Evaluation of the Search Module

The evaluation of the Search Module focuses on the precision, recall and diversity of the results. The top 30 results (results on the first page, as set in step 2.3) are suggested to be assessed for each aspect, as users are most likely to view an object on the first page of the search results.

Precision is the fraction of relevant instances among the retrieved instances [42]. The precision ($R(pre)$) of the Search Module is the number of objects in the list of search results that match the user's interests ($O(n)$) out of all the objects the module returns ($O(m)$).

$$R(pre) = \frac{O(n)}{O(m)} \tag{3}$$

The precision rate can be misleading since it can be affected by the number of items that meet the user's interests in the dataset. For example, if there is only one object in the dataset that meets the user's interests, the precision rate would be low inevitably; in such cases, precision would not be an appropriate indicator for the performance of the system. Another metric that should be considered in those cases is the recall rate. Recall is the fraction of relevant instances that were retrieved [42]. For the Search Module, the recall rate ($R(rec)$) would be the number of objects returned by the Search Module that match the user's interests ($O(n)$) out of all the objects that the user would find interesting in the database ($O(s)$), as calculated using the formula below. Due to the incomplete semantic annotations in the AS-NTUE-School-Art-Textbooks dataset, it would be difficult to calculate $O(s)$, but this metric would be applicable to other datasets with complete semantic descriptions.

$$R(rec) = \frac{O(n)}{O(s)} \tag{4}$$

The diversity of the results is assessed based on the similarity among the objects in the list of search results. The Node Similarity algorithm (unweighted) is used to compare any two objects among the first 30 objects returned by the Search Module based on the subjects and keywords they are associated with. The diversity rate ($D$) is then calculated using the formula shown below, where $S(avg)$ is the average similarity score, and the similarity scores vary from 0 to 1. The assumption here is that the higher the average similarity score is, the less diverse the results will be. The Cypher code and the procedures to measure the Node Similarity among the search results of an example search are presented in Appendix A.

$$D = 1 - S(avg) \tag{5}$$

(2)    Evaluation of the Recommendation Module

The evaluation of the Recommendation Module concentrates on precision and efficiency. This module is assumed to be used multiple times in a single search, and the user can check more than one batch of recommendations per round. This should be considered by the evaluation process.

The precision assessment for the Recommendation Module is mostly the same as the process for the Search Module, except for two things. First, the precision rates for each batch and round of results are calculated separately, and the change in the precision rate among different rounds and batches reveals the efficiency of the system. Second, when additional topics of interest are identified based on the user's viewing behaviours, only objects that match both the initially selected topics and the additional topics of interest inferred by the system are counted as effective results ($O(n)$).

The recall assessment is also necessary for the Recommendation Module, as the precision rate could be misleading for the same reason as for the Search module. The recall rate for the Recommendation Module is calculated in the same way as for the Search Module, except that the rate for each batch and round is measured separately, and the additional topics of interest are taken into account.

The Recommendation Module is designed to be used multiple times per search, and the more rounds the user interacts with the system, the more accurate the recommendations should be. This assumes that, while the user keeps viewing objects and interacting with the Recommendation Module, the system will have the chance to identify more topics of interest and also to assign appropriate values of relationship weight to the associated topics that show the degree of the user's interest in those topics. Evaluation of efficiency investigates the point that the user starts receiving more helpful results and whether the precision rate grows as the user views more rounds of results. The efficiency analysis is based on the precision rate of each batch and round. Increasing precision rates imply that the system is learning the user's interests based on their viewing behaviours and improving its performance accordingly.

## 4. Conclusions and Future Work

Semantic technologies have been widely applied to the cultural sector to improve information management, addressing the challenges posed by the diversity and heterogeneity of cultural data. In most cases, the target users of the tools and applications that have been developed using such technologies are administrators, professionals, and researchers. Less attention has been paid so far to the needs of the end users. This paper proposes recommendation methods and describes the architecture of a recommender system for cultural heritage collections based on graph databases and semantic technologies, in an attempt to exploit the semantic description of cultural collections for the benefit of the end users. The proposed system aims to assist users who search and browse cultural collections in cultural semantic portals with finding objects related to particular themes based on their semantic annotations. Exploiting the similar graph-based structures of RDF datasets and graph databases, it uses a powerful graph database system, Neo4j, to manage the available semantic metadata and implement various recommendation algorithms that account for the structure of the available semantic metadata, the preferences of the users, and their viewing behaviours. We developed and tested the recommendation methods on top of the Archive of the Art Textbooks of Elementary and Public Schools in the Japanese Colonial Period ("日帝殖民下台灣小公學校美術教科書暨影像數位典藏"), an RDF-based description of a cultural collection enriched with subject and keyword metadata.

One limitation of our study is the quality of the data that we used for developing and testing the recommendation methods. The example dataset is well-formatted and contains various semantic metadata, which we used to implement the recommendation algorithms. However, some of these annotations are incomplete or inconsistent, thus potentially affecting the performance of the algorithms. For example, some of the subjects or keywords are

provided in two languages (English and Chinese); this duplication of information can lead to problematic recommendations. We did not address this issue in the current study due to limited time but also because we wanted to keep the dataset intact.

The next steps of this work are mostly concerned with the further development of the proposed recommendation system and its evaluation. First, we want to test the algorithms with more semantic datasets and develop variations of the algorithms that will be able to utilise other types of semantic metadata. In addition to producing better recommendations, we also want our system to be easily adjustable and configurable to other types and formats of metadata. Another important next step is to implement a full version of the system based on the architecture we describe in this paper. We have already implemented parts of the system, such as the underlying graph database and the recommendation algorithms. We also want to develop the system interface and deploy the system on an existing semantic cultural portal and test it in a realistic context. A final but equally crucial step is to evaluate the system with real users. As described in Section 3, we have verified the soundness of the methods we developed using the test dataset. After developing the first full version of the system, we will evaluate the performance of the algorithm with real users using the process and the measures we describe in Section 3.4. We will also evaluate other aspects of the system such as its functionality, usability, reliability, performance, and scalability.

Although there is still a long way to go, we believe that the ideas and methods we present in this paper can demonstrate the potential of semantic and graph database technologies in exploiting existing semantic resources to provide intelligent end-user services in the cultural sector and inspire other digital humanities and data science researchers to further explore the potential of these technologies.

**Author Contributions:** Conceptualisation and methodology, J.L. and A.B.; software, validation, and resources, J.L.; writing—original draft preparation, J.L.; writing—review and editing, A.B. and J.L.; visualisation, J.L.; supervision, A.B. All authors have read and agreed to the published version of the manuscript.

**Funding:** This research received no external funding.

**Data Availability Statement:** Publicly available datasets were used in this study. This data can be found here: https://data.ascdc.tw/en/data.php.

**Conflicts of Interest:** The authors declare no conflict of interest.

## Appendix A

After the user selects and excludes topics, the Search Module starts running steps 2.1 and 2.2, as demonstrated in Section 3.3. Then, the evaluation follows the procedures below.

Instead of proceeding with step 2.3, a projection is created as the Cypher statements. The following code is similar to step 2.3, except that a property "test" is set for all the 30 object nodes, which are the first chunk of results returned by the Search Module to be evaluated, with the value of 1. The test property is set for node filtering in the following steps, and its value will not affect the evaluation results.

```
CALL gds.degree.stream(
'projection',
{relationshipWeightProperty: 'weight'}
)
YIELD nodeId, score
WITH gds.util.asNode(nodeId) AS object, score
ORDER BY score DESC LIMIT 30
SET object.test = 1
```

Then, a new projection is created to replace the one produced in step 2.2 in order to account for the test property, with four types of data yielded for further calculation listed below in the last Cypher statement.

```
CALL gds.graph.drop ('projection')

CALL gds.graph.project(
    'projection',
    {
    AscdcProvidedCHO: {properties: 'test'},
    Topic: {}
    },
    {
    subject: {},
    keywords: {}
    }
)
YIELD graphName, nodeCount, relationshipCount, projectMillis
```

A subgraph is created based on the projection we just created. The subgraph selects two types of nodes from the projection: the objects that are returned by the Search Module and all the topic nodes and relationships associated with these object nodes. Here, the test property is used for filtering the returned object nodes.

```
CALL gds.beta.graph.project.subgraph(
'sub_projection',
'projection',
'n:Topic OR (n:AscdcProvidedCHO AND n.test = 1)',
'*'
)
YIELD graphName, fromGraphName, nodeCount, relationshipCount
```

The Node Similarity algorithm is applied to the subgraph. Not applied in the filtered version as the Recommendation Module does in step 3.3, the algorithm compares each two object nodes in the subgraph based on the topic nodes they are connected to. Object URI and similarity scores are thus yielded.

```
CALL gds.nodeSimilarity.stream('sub_projection')
YIELD node1, node2, similarity
RETURN gds.util.asNode(node1).uri AS Object1, gds.util.asNode(node2).uri AS Object2,
similarity
ORDER BY similarity
```

The algorithm will return the URI for each pair of nodes and the similarity scores, which serve the evaluation of the diversity of results of the Search Module.

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
