# Peer review of "Towards a Semantics-Based Recommendation System for Cultural Heritage Collections"

_applsci, doi:10.3390/app13158907_

Round 1

Reviewer 1 Report

This is an interesting paper regarding the creation of semantics-based recommendation system for cultural heritage collections. The authors exploit the semantic metadata of the Archive of the Art Textbooks of Elementary and Public Schools in the Japanese Colonial Period dataset and with the employment of a graph database system they try toimprove the accuracy of the searches through the collection and the quality of the recommendations to the user.

The paper is well-structured giving in a great detail the steps that were followed to perform such a task. There are also examples and information about the queries that were used during the process.

My main concern about this work is the lack of evaluation. Along these lines of thought, I would recommend that the authors change the title of the paper to more closely reflect its content (additionally, the abstract and the intro should also be modified adequately). It is more of a hypothesis or a suggestion than the implementation of a real scenario. Without testing we cannot be sure that the proposed procedure is actually successful.

Author Response

Dear Reviewer 1,

thank you for taking the time to read and review our paper and for your useful comments. Please find attached a report describing how we addressed your and the other reviewers' comments in the revised version of the paper.

Best Regards,

Jiayu Lu and Antonis Bikakis

Reviewer 2 Report

This manuscript proposed a semantics-based recommendation system for the visitor's cultural portals' online collections to help them explore their online collections more efficiently. The authors tested their system with the Archive of the Art Textbooks of Elementary and Public Schools in the Japanese Colonial Period. This topic is interesting, and the manuscript is simply an understandable and well-structured presentation. However, several points in this manuscript should be substantially improved for publication.

Minors

Some of references are too old, I suggest to remove or update them. Besides, please consider to put more references relevant to the new approach to enhance RS, for example:

·         doi.org/10.1145/3488560.3501396

·         doi.org/10.1007/s41870-022-01011-x

·         doi.org/10.3390/app10124183

Sections 2.2 and 2.3 have the same title. Please check and update it.

The paper is comprised of many grammar mistakes. I suggest the authors carefully rewrite the paper or use the proofreading services before re-submission

Majors

The similarity checking is too high (46%). Please confirmed that it is a natural mistake and the authors must rephrase all similar sentences. I attached the checking results for reference (checked by Turnitin service)

I could not find out the proposed method (original idea). The author just described the way the search module working and the recommendations process (these modules has reference from previous works).

The experimental results were not presented,  for example, numerical results (e.g. evaluation results, etc). Instead, the authors just discuss the experiment setting and how to evaluate it. The reviewer suggests adding the section "experimental results" with the details results, and explanation, showing the performance of the proposed approach in comparison with the other recent approach.

Minor editing of English language required

Author Response

Dear Reviewer 2,

thank you for taking the time to read and review our paper and for your useful comments. Please find attached a report describing how we addressed your and the other reviewers' comments in the revised version of the paper.

Best Regards,

Jiayu Lu and Antonis Bikakis

Reviewer 3 Report

Have to check the entire manuscript for clarity and readability. Reorganizing entire manuscript is necessary

Extensive English proof reading is essential

Author Response

Dear Reviewer 3,

thank you for taking the time to read and review our paper and for your useful comments. Please find attached a report describing how we addressed your and the other reviewers' comments in the revised version of the paper.

Best Regards,

Jiayu Lu and Antonis Bikakis

Round 2

Reviewer 2 Report

This revised version is good enough to consider for publication.

Minor editing is required